# Body Size Modulates the Impact of the Dispersive Patch Position During Radiofrequency Cardiac Ablation

**DOI:** 10.3390/bioengineering12101017

**Published:** 2025-09-24

**Authors:** Ramiro M. Irastorza, Enrique Berjano

**Affiliations:** 1Instituto de Física de Líquidos y Sistemas Biológicos (CONICET), La Plata B1900, Argentina; rirastorza@iflysib.unlp.edu.ar; 2Departamento de Ingeniería Mecánica, Facultad Regional La Plata, Universidad Tecnológica Nacional, La Plata B1900, Argentina; 3BioMIT, Department of Electronic Engineering, Universitat Politècnica de València, Building 7F, Camino de Vera, 46022 Valencia, Spain

**Keywords:** body size, cardiac ablation, dispersive patch, radiofrequency ablation, ventricular ablation

## Abstract

(1) Background: In the context of cardiac radiofrequency (RF) ablation, it has been proposed that positioning the dispersive patch (DP) concordantly with the orientation of the ablation electrode may enhance lesion size. The objective of this study is to investigate how individual body size may modulate the extent of this effect. (2) Methods: Three computational models representing different body sizes were developed. An irrigated catheter ablation was simulated by delivering a 30 W pulse for 30 s to the endocardial surface of the anterior wall. Lesion sizes were then compared between two configurations of the dispersive patch (DP): an anterior (concordant) position and a posterior (discordant) position. (3) Results: Lesion size was consistently and significantly greater with concordant DP positioning compared to discordant positioning. Moreover, the magnitude of this difference decreased significantly with increasing body size, ranging from 0.65 ± 0.08 mm in the 35 kg swine model to 0.51 ± 0.06 mm in the human model. (4) Conclusions: Body size has a modest influence on the effect of dispersive patch positioning on RF lesion size. The potential advantage of a concordant DP configuration may be more significant in individuals with smaller body volume.

## 1. Introduction

Radiofrequency (RF) catheter ablation (RFCA) has become an essential therapeutic strategy for the management of ventricular arrhythmias, particularly in patients with structural heart disease and recurrent ventricular tachycardia (VT) [1]. RFCA operates by delivering alternating current at a frequency of approximately 500 kHz between an active electrode (ablation electrode), positioned at the distal tip of the catheter, and a dispersive electrode (dispersive patch, DP) placed on the patient’s skin (see Figure 1). Despite significant advances in mapping technologies and energy delivery systems, the efficacy of ablation procedures remains limited in part by the inability to consistently achieve transmural lesions, especially in the presence of intramural or epicardial arrhythmogenic substrates. The complex three-dimensional architecture of the ventricular myocardium, often altered by fibrosis or scarring, poses a substantial challenge for effective lesion formation. Consequently, there is a growing need to refine ablation techniques that can reliably produce sufficiently deep to eradicate the arrhythmic focus and reduce the risk of recurrence. This necessitates not only technological improvements but also a better understanding of lesion dynamics, energy–tissue interactions, and real-time lesion assessment during the procedure.

In the context of RFCA within the ventricular myocardium, a recent approach proposes positioning the DP on the body surface in concordance with the “direction of RF ablation” (i.e., the direction in which the ablation electrode points), which, according to a recent study, would allow the lesion depth to increase by 20% [2] (see Figure 1). The underlying rationale is that orienting the DP in the same direction as the RF catheter (concordant position) facilitates more efficient current flow toward the myocardial target, thereby reducing current dissipation into the blood pool. However, clinical, preclinical, and computational investigations into this technique have yielded inconsistent findings to date.

The only two clinical evidence on the effect of the DP position (i.e., clinical trials where different positions were evaluated) are the pilot studies of Nath et al. [3] and Futyma et al. [4]. While the first one reported no differences in baseline impedance between interscapular and left thigh positions (101 ± 10 vs. 101 ± 9 Ω) in 20 patients [3], the second one reported a significant difference between anterior and posterior position (134 ± 7 vs. 122 ± 8 Ω, *p* < 0.001) in 64 patients [4]. It is important to highlight that Nath et al. conducted an intra-individual comparison (using Student’s paired *t* test), since the impedance measurement for each DP position was made on the same subject. In contrast, Futyma et al. compared retrospectively the impedance measurements of 40 patients with posterior position and 22 patients in anterior position. The type of RF catheter and the ablation procedure were also different in both studies, making comparison between them almost impossible. Unfortunately, clinical studies only provide information on the effect of DP position on baseline impedance, and not on lesion size, unlike in vivo models.

In this regard, and to the best of our knowledge, only two preclinical studies have been conducted to evaluate the effect of the DP position on RF lesion size. Jain et al. [5] compared the size of 27 pairs of RF lesions conducted on left and right ventricle in six adult female sheep (56 ± 2 kg). Each pair of lesions were made side by side in order to minimize variability in lesion size due to anatomic location, one with a concordant position of the DP and the other with a discordant position, and they were then compared using a paired Student’s *t*-test. Although no significant differences were found in the baseline impedances between concordant and discordant DP positions (113.1 ± 31.8 vs. 113.2 ± 31.9 Ω), they reported significantly deeper and larger lesion sizes with the concordant position: 5.8 ± 0.8 vs. 4.6 ± 1.0 mm in depth, and 9.3 ± 1.9 vs. 7.7 ± 1.9 mm in maximum width (*p* < 0.001) [5]. More recently, Venkateswaran et al. [2] conducted a comparative analysis of 38 RF lesions with concordant DP position and 39 lesions with discordant position. All lesions were created in the left ventricle of ten farm swine (36 ± 11 kg). Using unpaired *t*-tests, they did not find a statistically significant difference in baseline impedance (111.3 ± 7.5 vs. 110.9 ± 7.3 Ω) between concordant and discordant positions, but deeper (7.4 ± 1.8 vs. 6.1 ± 1.6 mm) and wider (10.1 ± 1.9 vs. 9.2 ± 1.5 mm) lesions with the concordant position. The conclusions were hence similar in both pre-clinical studies: the concordant DP position produces a statistically significant increase in lesion depth of up to 1.2–1.3 mm [2,5] compared to the discordant position, even without differences in baseline impedance.

One of the limitations of pre-clinical experimental studies aimed at evaluating the effect of DP position on RF lesion size is the use of animals with a small body size, which, as already suggested by Venkateswaran et al. [2], may modulate the impact of DP positioning. From a bioelectrical point of view, intuition suggests that the smaller the body size, the stronger the “RF current redirection effect” will be when the DP position is changed. Addressing this issue was precisely the objective of the present study, which employed an in silico model to explore the effect of the body size on the potential benefit of using the DP in a position concordant with the ablation electrode in order to increase the RF lesion size during RFCA in the ventricle.

In silico modeling, when applied to biomedical sciences—and particularly to the development and assessment of medical devices—provides a powerful framework for the systematic investigation of biophysical phenomena. This approach offers significant advantages over traditional experimental methods, including reduced cost and time requirements, enhanced reproducibility, and the ability to isolate and control individual parameters. Furthermore, computational models enable the quantification of physical variables that are otherwise difficult or impossible to measure in vivo or in vitro, thereby supporting more comprehensive and mechanistic insights into complex biological systems.

## 2. Materials and Methods

### 2.1. Geometry Description

Three-dimensional models were built to simulate ventricular RFCA. The model geometry has been previously described in [6], and includes the heart (modeled as a sphere), a fragment of the lungs, bony structures (spine and sternum) and subcutaneous adipose tissue (the skin is ignored due its negligible thickness). The torso comprised a mixture of skeletal muscle and fat, as observed in CT-images [6]. In order to assess the impact of the body size, three models were built based on the described geometry, but scaled according to the anteroposterior length, as shown in Figure 2: (1) human model (anteroposterior length of 27 cm), (2) 45 kg large swine model (anteroposterior length of 24 cm), and (3) 35 kg small swine model (anteroposterior length of 19 cm). The lengths of the two animal models were provided by our veterinary collaborators. All internal anatomical structures were proportionally resized according to the anteroposterior length, except for the RF electrode and dispersive patch (DP), whose dimensions remained constant. Additionally, ventricular wall thickness and electrode insertion depth were independently varied across a range of clinically relevant values—4 to 7 mm for wall thickness, and 0.3 to 0.7 mm for insertion depth—resulting in 20 distinct simulations for each body size. This simulations plan was designed to account for the physiological and anatomical variability encountered in pre-clinical and clinical scenarios.

The RF catheter comprised a metal electrode (7Fr, 3.5 mm) and a fragment of plastic tube. The DP was assumed to be a 7 cm radius disk, with a contact area of 154 cm^2^ (which is a value very similar to the commercially available DPs). Two positions were evaluated: anterior (concordant) and posterior (discordant).

### 2.2. Governing Equations and Boundary Conditions

An electro-thermal coupled problem was solved by the Finite Element Method, in particular using the FEniCS numerical software (version 2019.1.0) and Gmsh mesher (version 4.12.1). The electrical problem was solved from Laplace’s equation, in order to obtain the spatial distribution of electrical voltage *φ*(1)∇·(σ∇φ)=0
where *σ* is the electrical conductivity (S/m). The electric field ***E*** distribution was then obtained from *φ* using(2)E=−∇φ

The RF power density deposited in the tissue *Q_RF_* (W/m^3^) was computed as(3)$$Q_{\mathrm{RF}}=\sigma {\left|E\right|}^{2}$$

This value was coupled with the Bioheat Equation to calculate the temperature distribution in tissue and electrode:(4)$$\rho c\frac{\partial T}{\partial t}=\nabla \cdot (k\cdot \nabla T)-Q_{p}+Q_{m}+Q_{\mathrm{RF}}$$
where *ρ* is density (kg/m^3^), *c* specific heat (J/kg∙K), T temperature (°C), *t* time (s), *k* thermal conductivity (W/m∙K), *Q_p_* the heat loss caused by blood perfusion (W/m^3^) and *Q_m_* the metabolic heat generation (W/m^3^). The last two terms were ignored as being negligible compared to the others [7].

An electrical boundary condition of 0 V was set on the torso surface corresponding to the position of the DP, while a boundary condition of 0 A was set on the rest of the torso surface. The RF current through the active electrode was modulated during the ablation to keep power constant. An RF pulse of 30 W during 30 s was simulated. The thermal problem was not solved either in the cardiac chamber containing blood; a thermal transfer coefficient of 4500 W/m^2^·K [8] was instead used to model the cooling effect of the circulating blood. Initial temperature was set at body temperature (37 °C). The electrode irrigation was modeled by fixing a value of 45 °C in the cylindrical zone of the electrode tip, and leaving the semispherical tip free, mimicking a multi-hole electrode assuming that irrigation occupies almost the entire surface of the electrode [6]. The lesion size was quantified using the 55 °C isotherm [9]. Depth, surface width, and maximum width (regardless of location) were systematically analyzed.

The 2D axisymmetric electro-thermal coupled problem was solved by the Finite Element Method and we used Newton’s method to linearize the residual equation. A backward Euler scheme was adopted for the transient integration and the relative tolerance was set at 0.2%. The fixed time-step parameter was set at 0.001 s. The sufficiency of the degree of meshing and the time step in the transient simulation were verified by a sensitivity analysis using the lesion size as the control parameter and a relative variation in less than 0.5% as a criterion. For example, using the small-swine model as a test, it initially contained approximately 6350 elements. At 10 s, the maximum width of the coagulation zone (55 °C isotherm) was 8.18 mm. With 10,613 elements, the coagulation zone measured 8.16 mm, i.e., a difference of 0.02 mm.

### 2.3. Material Properties

Table 1 shows the characteristics of the materials used in the model. The tissue electrical and thermal properties were taken from the IT’IS Foundation database [10]. The values for the lung were the mean of the inflated and deflated values. Those for bone (spine and sternum) were the mean of the cortical and trabecular bones. Those for the tissue surrounding the organs were a mixture of 61% infiltrated fat and 39% muscle, the average values found after an analysis of the CT-scans of 20 patients [6]. The 6 mm outer layer (SAT: subcutaneous adipose tissue) was assumed to be subcutaneous fat (see Figure 1). The electrical conductivity of tissue was assumed to increase by +1.5/°C. The catheter materials (metal and plastic) were taken from [11].

### 2.4. Statistics

This study used a physics-based mechanistic model. While the three body sizes were fixed in the model geometry, variability was introduced into the model by considering a range of cardiac wall thickness (4–8 mm) and of insertion depths of electrode (0.3–0.7 mm). All case permutations with respect to the above variables involved a total of 60 simulations (20 for each body size). The comparison of results between DP positions and body sizes was performed using a non-parametric test (Mann–Whitney U test). Statistical significance was assumed when the *p*-value (*p*) was lower than 0.05.

## 3. Results

No simulation reached 100 °C in the tissue, nor was there any roll-off (rapid increase in impedance). Appendix A includes the Excel file with all the outcomes of the computer simulations. Table 2 shows the lesion sizes for each body size and DP position, while Figure 3 shows the significant differences in lesion sizes for each DP position and body size. In general, it is observed that the effect of body size on lesion size is much more pronounced in the case of concordant position, which coincides in this model with the smallest distance between the ablation electrode and DP. Figure 4 illustrates the differences in lesion depth obtained with concordant vs. discordant DP positioning across the three body size models. In all cases, lesion depth was significantly greater with concordant positioning compared to discordant positioning: 0.65 ± 0.08 mm in the 35 kg swine model, 0.58 ± 0.07 mm in the 45 kg swine model, and 0.51 ± 0.06 mm in the human model. These findings suggest a consistent trend of decreasing lesion size with increasing body size, which was also observed in the lesion width: the concordant position implied a statistically significant increase respect to the discordant position, from 1.06 ± 0.06 mm of maximum width and 0.67 ± 0.04 mm of surface width in the 35 kg swine model, to 0.84 ± 0.02 mm of maximum width and 0.58 ± 0.01 mm of surface width in the human model.

Table 3 shows the baseline impedance and RF current computed at the mid-point of the RF pulse (i.e., at 15 s) for each body size and DP position, while Figure 5 shows the statistically significant differences observed in these parameters. Once more, overall, it is observed that the effect of body size on lesion size is much more pronounced in the case of concordant position, which coincides in this model with the smallest distance between the ablation electrode and DP. Figure 6 illustrates the differences in baseline impedance obtained with concordant vs. discordant DP positioning across the three body size models. In all cases, baseline impedance was significantly greater with discordant positioning compared to concordant positioning: 20.6 ± 0.1 Ω for 35 kg swine model, 18.4 ± 0.1 Ω for 45 kg swine model, and 18.1 ± 0.1 Ω for human model.

## 4. Discussion

### 4.1. Effect of the DP Position

Assuming that a specific position of the DP on the patient’s skin during monopolar RF ablation will impact the lesion size around the active electrode thanks to a local modulation of the current density distribution is an old but little-studied topic. Beyond ensuring that the DP is located in an area that ensures good contact to avoid collateral thermal damage such as skin burns [12], collective analysis of studies assessing the influence of specific DP positioning on RF lesion size has yielded inconclusive findings.

With regard to computational modeling studies, and to the best of our knowledge, the first study was conducted by Shahidi and Savard [13], who modeled the body as a structure composed of concentric spherical layers extending from the cardiac chamber (with the RF catheter positioned on the endocardium) to the skin surface, including the cardiac wall and torso, resulting in a model with a 20 cm total diameter. They found that a discordant DP position resulted in only a 0.2 mm increase in lesion depth.

Later, Jain et al. [5] compared the RF lesion sizes for both DP positions using a limited-domain computer model with 8 cm per side. This geometry placed the 0 V boundary condition, representing the DP position, only 4 cm away from the RF electrode, which is unrealistic and possibly exaggerates the modulation of the spatial distribution of current around the active electrode. Their computer simulations showed deeper (6.6 vs. 5.9 mm) and wider (9.0 vs. 8.6 mm) lesions with the concordant DP position [5]. Recently, we built a full torso model based on a spherical heart and simulated RF ablations on the anterior ventricular wall [6]. We found a minimal increase in RF lesion depth—only 0.6 mm—when the DP was positioned concordantly with the ablation catheter. Finally, the recent study by Anees et al. [14] employed a 3D model whose geometry was derived from medical imaging to evaluate how RF power dissipates across different regions of the torso depending on the DP position. However, the study did not solve a coupled electro-thermal problem, and therefore, it was unable to capture key information regarding the heating dynamics or draw conclusions about RF lesion size.

While computational modeling is a valuable tool for investigating this topic—enabling precise control over parameters that are otherwise unmanageable in preclinical experiments or clinical trials—it is important to recognize that current state-of-the-art models may not fully replicate the conditions present during RFCA. Indeed, it is reasonable to assume that the spatial distribution of RF current around the active electrode, as well as its conduction path toward the DP, is highly dependent on the anatomical geometry of the intervening tissues, which in turn is strongly patient-specific.

Therefore, acknowledging the limitations inherent to any methodological approach used to study this subject, it is essential to integrate all available evidence to progressively advance our understanding. This need is particularly pressing given that current RFCA protocols are based on constant power delivery, meaning that lesion size is primarily determined by the power dissipated in the ‘local impedance’—i.e., the impedance of the tissue surrounding the ablation electrode. However, repositioning the dispersive patch (DP) can also influence the ‘remote impedance’, thereby indirectly modifying the power delivered to the active electrode and, consequently, affecting lesion size [15]. Moreover, under specific conditions, it is conceivable that repositioning the DP could directly impact the ‘local impedance’ by modifying the current distribution surrounding the active electrode.

Thus, findings from previous studies suggest that the increase in RF lesion size observed with a concordant DP position may result from two interrelated mechanisms, whose relative contributions likely vary depending on individual patient anatomy: (1) a reduction in baseline impedance, leading to an increase in total delivered RF current; and (2) a modulation of the spatial distribution of current density that favors greater energy deposition within the myocardial tissue while reducing current dissipation into the intracavitary blood. In the following paragraphs, we will examine these two mechanisms in detail.

Firstly, a change in baseline impedance can be attributed to alterations in its ‘remote’ component—that is, the component that does not directly affect the current distribution around the active electrode. For instance, an increased distance between the active electrode and the DP, or a higher concentration of adipose tissue within the torso, may elevate the ‘remote’ component of baseline impedance. This, in turn, leads to greater power dissipation in the remote region and consequently reduces the power delivered to the ‘local’ zone surrounding the active electrode, resulting in smaller lesion size. Essentially, any increase in baseline impedance corresponds to a decrease in total current delivered under constant power ablation mode, which reduces current density near the active electrode and thus lesion size. This phenomenon has been observed both in our current study and in previous investigations [6,16], where larger lesion sizes associated with the concordant DP position were correlated with reduced baseline impedance and increased total delivered current.

The second mechanism could involve a direct modulation of the spatial distribution of RF current around the active electrode, as suggested in previous studies [2,4,5]. Specifically, this refers to the tendency of RF current ‘leaving’ the ablation electrode to preferentially flow toward the DP position. Notably, this phenomenon has been proposed to occur without significant changes in total impedance (*Z_T_*) between different DP positions, and thus without substantial variation in delivered RF current (*I_RF_*), given that the applied power (*P_T_*) remains constant. Figure 7 illustrates the proposed mechanism using a simple lumped electrical model to support the hypothesis, where *Z_R_* represents the ‘remote’ component of impedance and *Z_L_* the ‘local’ component. Although the distinction between remote and local impedance is somewhat arbitrary, in this context, ‘local’ is defined as the region within ∼1 cm of the active electrode. It is important to note that the model depicted in Figure 7 does not differentiate between the impedances of intracavitary blood and myocardium, as was performed in [17]; instead, both are combined into the single local impedance component, *Z_L_*. Using this model, the power involved in the thermal lesion formation (*P_L_*) is expressed as(5)$$P_{L}=I_{RF}^{2}\cdot Z_{L}$$

On the other hand, the lost power (*P_R_*), defined as the power dissipated in *Z_R_* and not contributing to thermal lesion formation, is given by(6)$$P_{R}=I_{RF}^{2}\cdot Z_{R}$$

If we assume that the baseline impedance remains constant when altering the position of the DP, then(7)$$Z_{R\text{-}dis}+Z_{L\text{-}dis}=Z_{R\text{-}con}+Z_{L\text{-}con}$$

Moreover, assuming that *I_RF_* remains unchanged due to constant applied power, then(8)$$I_{RF}=\sqrt{\frac{P_{T}}{Z_{R\text{-}dis}+Z_{L\text{-}dis}}}=\sqrt{\frac{P_{T}}{Z_{R\text{-}con}+Z_{L\text{-}con}}}$$
which implies that when a larger lesion is produced using the concordant DP position, it is due to an increase in *Z_L-con_* accompanied by a proportional decrease in *Z_R-con_*. And conversely, when a smaller lesion occurs with the discordant position, it results from a decrease in *Z_L-dis_* coupled with a proportional increase in *Z_R-dis_*.

The actual scenario is likely more complex, with both aforementioned mechanisms occurring simultaneously during DP repositioning. Nevertheless, we believe that predicting the exact outcome is challenging, as it is highly dependent on individual patient-specific factors.

### 4.2. Effect of the Body Size

From a bioelectrical standpoint, it is intuitive to hypothesize that as the distance between the two electrodes (active and DP) decreases, the distribution of dissipated power around the active electrode becomes increasingly dependent on the position of the dispersive electrode (DP). This circumstance may arise either when the ablation site is located near the body surface (e.g., during ablation of the anterior ventricular wall) or in patients with smaller body size. This idea is supported by our computational findings, which demonstrate a progressive increase in the difference in lesion depth between concordant and discordant DP positions—from 0.51 mm to 0.65 mm—as body size decreases. In our case, this observed increase in lesion depth may be explained by the greater total RF current delivered with concordant DP positioning, which is associated with a reduction in baseline impedance.

It is noteworthy that in the case of a discordant DP position—i.e., when the active electrode is positioned farther from the DP—no significant differences in baseline impedance or RF current were observed among the three body sizes. In contrast, both baseline impedance and RF current differed significantly among the three body sizes in the case of a concordant DP position—that is, when the active and dispersive electrodes were in closer proximity. Interestingly, this behavior was not directly reflected in lesion depth, as differences were observed between the two extreme body sizes (human model vs. 35 kg swine model). However, these differences were more pronounced in the concordant DP position (*p* < 0.001) compared to the discordant position (*p* = 0.037).

An early pilot trial by Borganelli et al. [18], involving 27 patients undergoing RFCA, reported a moderate correlation (R^2^ = 0.44, *p* = 0.0001) between body surface area (BSA) and baseline impedance (ranging from 77 to 132 Ω). At that time, control over the contact force exerted by the ablation electrode on the endocardium was not yet available. Therefore, the portion of the variability not explained by BSA could be attributed to dispersion in both the ‘local impedance’—due to variations in electrode–tissue contact—and the ‘remote impedance’—resulting from differences in tissue composition along the current path to the DP, primarily the amount of adipose tissue infiltrating skeletal muscle and subcutaneous regions.

### 4.3. Clinical Impact

Considering all the evidence to date together with our computational results, we can conclude that a reduction in baseline impedance will generally allow more RF current to be delivered, resulting in larger lesion size. This reduction in baseline impedance can be achieved in several ways (e.g., using two patches instead of one [3]), with positioning the DP closer to the ablation electrode being one possible approach. This potential effect is modulated by body size, being more pronounced in the case of smaller body sizes.

In terms of clinical impact, the differences in lesion depth observed in the computational results when both the DP position and body size changed were small (<1 mm). Although these figures could be considered insignificant, we believe that nothing should be disregarded when it comes to ablating deep arrhythmogenic foci, and in fact our findings suggest to the electrophysiologist that repositioning the DP will have an impact on the lesion depth to the extent that it implies a decrease in the baseline impedance (associated with a reduction in the distance between the ablation electrode and the DP), and that such a maneuver is expected to have a greater impact in the case of patients with smaller body size.

### 4.4. Limitations

General limitations of computational modeling have been extensively discussed in previous studies [15,16]. In our specific case, the anatomical geometry was uniformly scaled while preserving relative proportions, and the most relevant organs were included with appropriate dimensions, as well as representative electrical and thermal properties. However, average values were assigned to certain tissue properties, and variations such as localized fat deposits or differences in external body contour could alter current distribution pathways, potentially influencing the ablation outcome. Additionally, when using a small animal model, the application of a standard-sized DP may not achieve full surface contact, which could affect baseline impedance and the resulting subcutaneous current distribution.

Although the model was derived from CT medical images, it incorporated a simplified geometry that included only the most relevant organs and tissues. The extent to which incorporating a higher level of anatomical detail might affect the results remains uncertain. In this context, it is noteworthy that other computational studies did not report differences in baseline impedance between DP positions, whereas our simulations did reveal such differences, which may have directly contributed to the observed variations in lesion size. Despite these discrepancies, our findings confirm that body size exerts a modest but measurable influence on lesion size. This factor may be relevant in clinical decision-making when considering DP repositioning to enhance lesion formation, and should also be taken into account when interpreting results from preclinical studies conducted in small- to medium-sized animal models.

## 5. Conclusions

Our findings suggest that body size has a modest influence on the effect of dispersive patch positioning on RF lesion size. Specifically, the potential clinical advantage of a concordant DP configuration (compared to a discordant one) may be more significant in individuals with smaller body volume. This can be partially due to a greater reduction in baseline impedance associated with DP repositioning, which facilitates increased RF current delivery through the myocardium, thereby enhancing lesion size.

## Figures and Tables

**Figure 1 bioengineering-12-01017-f001:**
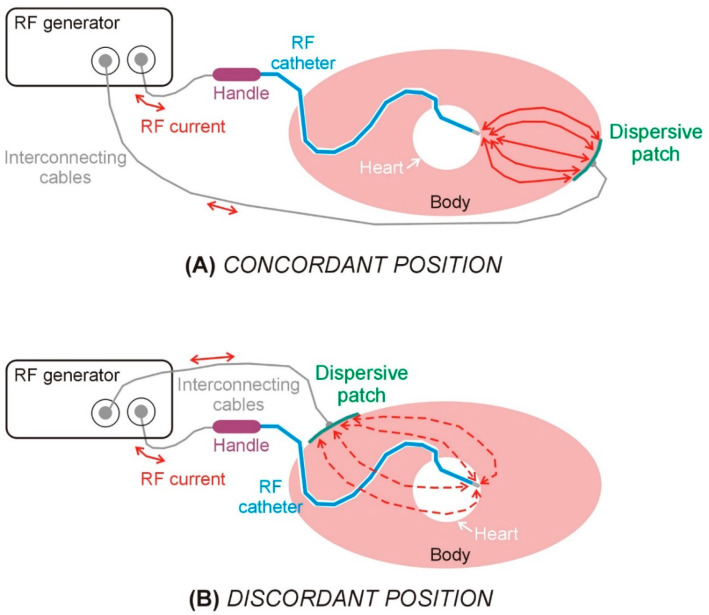
Schematic views of the elements involved in a radiofrequency catheter ablation in the case of dispersive patch in concordant (**A**) and discordant (**B**) positions.

**Figure 2 bioengineering-12-01017-f002:**
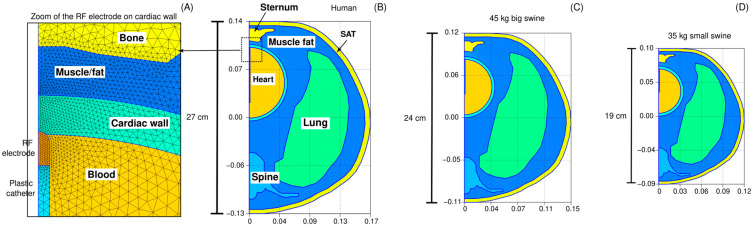
Model geometries. Panel (**A**) provides a magnified view of the finite element mesh in the vicinity of the active electrode. Three anatomical models scaled to match the anteroposterior length of each geometry: (**B**) a human model (27 cm), (**C**) a 45 kg large swine model (24 cm), and (**D**) a 35 kg small swine model (19 cm). SAT: Subcutaneous adipose tissue.

**Figure 3 bioengineering-12-01017-f003:**
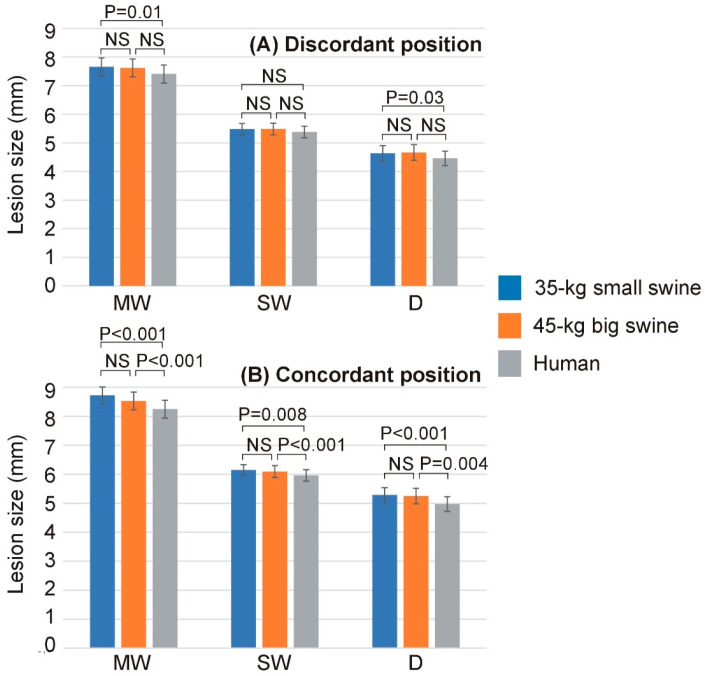
Lesion dimensions (MW: maximum width; SW: surface width; D: depth) across the three body sizes (35 kg small swine, 45 kg large swine, and human) and the two dispersive patch (DP) positions ((**A**): discordant; (**B**): concordant). Error bars represent the standard deviation. NS: Not significant difference.

**Figure 4 bioengineering-12-01017-f004:**
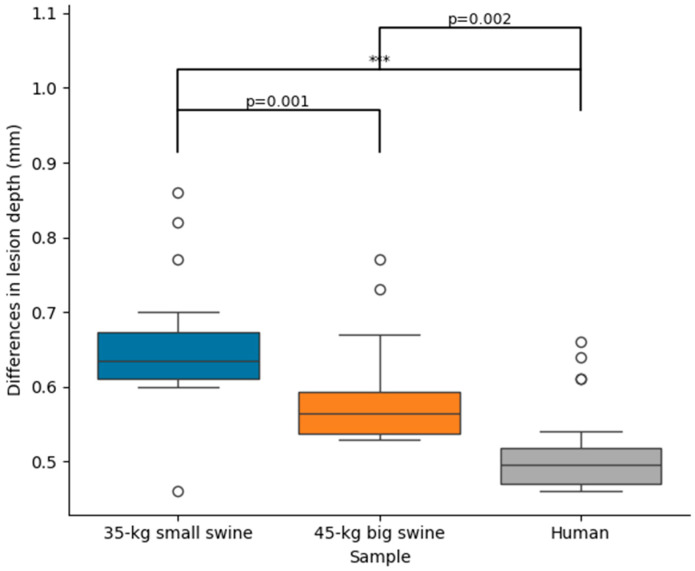
Differences in lesion depth between concordant and discordant DP position across the three body sizes (35 kg small swine, 45 kg large swine, and human). Significant differences (***, *p* < 0.001). The circles represent outliers, while the whiskers extend to the most extreme values within 1.5 times the interquartile range.

**Figure 5 bioengineering-12-01017-f005:**
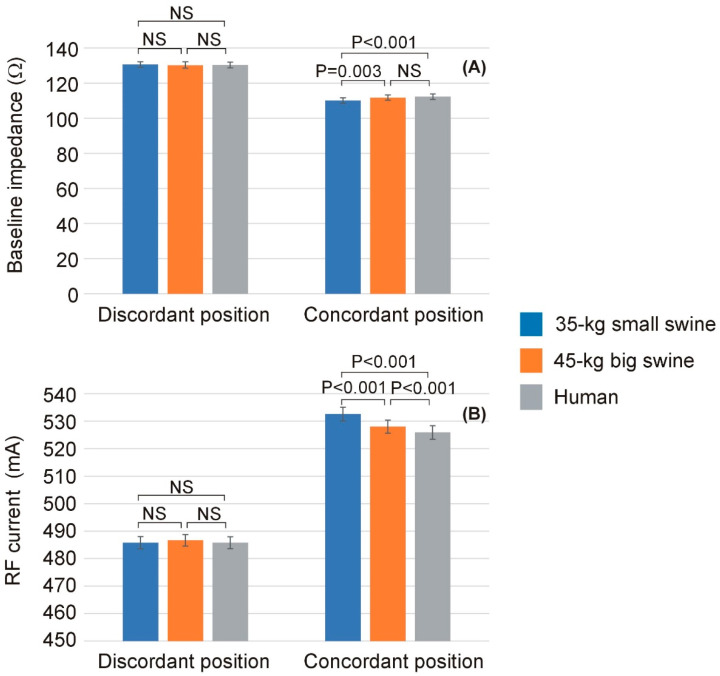
Baseline impedance (**A**) and RF current (**B**) computed at the mid-point of the RF pulse (i.e., at 15 s) across the three body sizes (35 kg small swine, 45 kg large swine, and human) and the two dispersive patch (DP) positions (discordant and concordant). Error bars represent the standard deviation. NS: Not significant difference.

**Figure 6 bioengineering-12-01017-f006:**
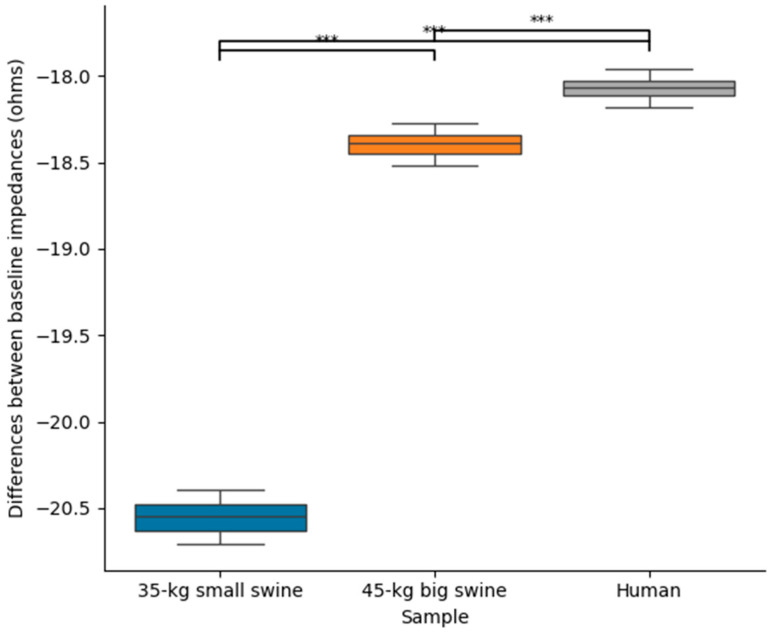
Differences between baseline impedances between concordant and discordant DP position across the three body sizes (35 kg small swine, 45 kg large swine, and human). Significant differences (***, *p* < 0.001).

**Figure 7 bioengineering-12-01017-f007:**
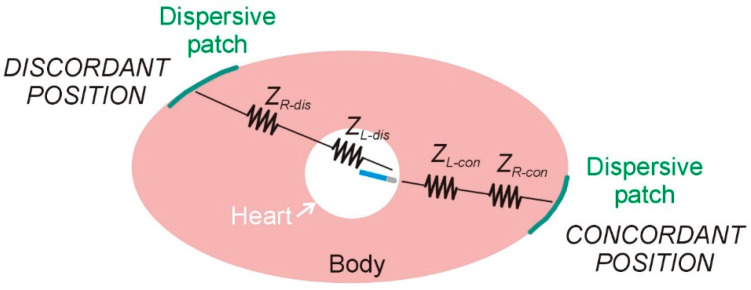
Lumped element electrical model based on two resistors (local and remote impedances) for radiofrequency catheter ablation under two conditions of dispersive patch position: concordant and discordant.

**Table 1 bioengineering-12-01017-t001:** Thermal and electrical characteristics of the elements of the model [10,11].

Element/Material	*σ* (S/m)	*k* (W/m·K)	*ρ* (kg/m^3^)	*c* (J/kg·K)
Electrode/Platinum–Iridium	4.6 × 10^6^	71	21,500	132
Catheter/Polyurethane	10^−5^	23	1440	1050
Cardiac wall/Myocardium	0.281	0.56	1081	3686
Cardiac chamber/Blood	0.748			
Muscle	0.446	0.49	1090	3421
Subcutaneous fat (infiltrated fat)	0.0438	0.21	911	2348
Lungs	0.215	0.39	722	3886
Spine, sternum/bone	0.055	0.315	1543	1793

*σ*: electrical conductivity (at 500 kHz); *k*: thermal conductivity; *ρ*: density; *c*: specific heat (all assessed at 37 °C in case of tissue and blood).

**Table 2 bioengineering-12-01017-t002:** Lesion sizes (mm) for each body size and dispersive patch (DP) position.

	Posterior DP (Discordant)			Anterior DP (Concordant)		
	MW	SW	D	MW	SW	D
Small swine	7.65 ± 0.32	5.48 ± 0.20	4.63 ± 0.27	8.71 ± 0.30	6.15 ± 0.19	5.28 ± 0.25
Big swine	7.62 ± 0.31	5.48 ± 0.20	4.66 ± 0.28	8.53 ± 0.31	6.09 ± 0.20	5.25 ± 0.27
Human	7.41 ± 0.31	5.38 ± 0.25	4.46 ± 0.25	8.25 ± 0.31	5.96 ± 0.20	4.97 ± 0.25

MW: Maximum width; SW: Surface width; D: Depth.

**Table 3 bioengineering-12-01017-t003:** Baseline impedance and RF current for each body size and dispersive patch (DP) position.

	Posterior DP (Discordant)		Anterior DP (Concordant)	
	Baseline Impedance (Ω)	RF Current (mA)	Baseline Impedance (Ω)	RF Current (mA)
Small swine	131 ± 2	486 ± 2	110 ± 2	533 ± 3
Big swine	130 ± 2	487 ± 2	112 ± 2	528 ± 2
Human	130 ± 2	486 ± 2	112 ± 2	526 ± 3

## Data Availability

The raw data supporting the conclusions of this article are included in the Appendix A.

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
