# Peer review of "Body Size Modulates the Impact of the Dispersive Patch Position During Radiofrequency Cardiac Ablation"

_bioengineering, 2025, doi:10.3390/bioengineering12101017_

Round 1

Reviewer 1 Report

Comments and Suggestions for Authors

This is a well written, interesting manuscript. Attention to the following should improve it.
Minor comments
1.    On line 85, change: “did no find statistical difference” to “did not find a statistically significant difference”.
2.    On line 88, change: “implies” to “produces”.
3.    On line 96, it would be helpful to explain that In biology and other experimental sciences, an in-silico experiment is performed on a computer or via computer simulation software.
4.    On line 119, change: “comprised” to “was comprised”.
Major Comments
1.    For readers unfamiliar with box plots (Figure 4), the terminology is clarified by McLeod S. Reading A Box And Whisker Plot. Available at: https://www.simplypsychology.org/boxplots.html
2.    Figure 5 suggests that baseline impedance was lower in the concordant alignment and that current was higher in the concordant position during RF delivery. Lines 223-225 seem to say the opposite about impedence. Please address this apparent discrepancy. 
3.    The figure 6 legend mentions circles, but none are present in the figure. Please address this discrepancy. 
4.    On lines 340-342, you have written: From a bioelectrical perspective, it is intuitive to hypothesize that the possible “RF current redirection effect” resulting from variations in the DP position would be more pronounced when the active and dispersive electrodes are in closer proximity”. Are you describing dispersion of energy encircling the heart away from the DP? If not, this hypothesis seems illogical. Please clarify.

Author Response

Thank you very much for the prompt and thorough review. We have carefully addressed all comments and incorporated the corresponding revisions in the updated version of the manuscript.

COMMENT #1: On line 85, change: “did no find statistical difference” to “did not find a statistically significant difference”.”

Change in manuscript: Done.

COMMENT #2: On line 88, change: “implies” to “produces”

Change in manuscript: Done.

COMMENT #3: On line 96, it would be helpful to explain that in biology and other experimental sciences, an in-silico experiment is performed on a computer or via computer simulation software.

Change in manuscript: A brief comment has been added about this point.

COMMENT #4: On line 119, change: “comprised” to “was comprised”.”

Change in manuscript: Done.

COMMENT #5: For readers unfamiliar with box plots (Figure 4), the terminology is clarified by McLeod S. Reading A Box And Whisker Plot. Available at: https://www.simplypsychology.org/boxplots.html”

Response: It is challenging to cite this website as a formal reference. The authors assume that readers seeking to understand the interpretation of this type of graph will be able to readily access appropriate sources through the scientific literature or reliable online resources.

COMMENT #6: Figure 5 suggests that baseline impedance was lower in the concordant alignment and that current was higher in the concordant position during RF delivery. Lines 223-225 seem to say the opposite about impedance. Please address this apparent discrepancy.”

Response: Thank you for pointing this out—this was indeed an error. The impedance in the discordant position was consistently higher, which accounts for the negative values observed in the differences shown in Fig. 6.

COMMENT #7: The figure 6 legend mentions circles, but none are present in the figure. Please address this discrepancy.”

Response: The error occurred because part of the legend from Fig. 4 was inadvertently copied into the legend of Fig. 6. Unlike Fig. 4, Fig. 6 does not include any outliers.

COMMENT #8: On lines 340-342, you have written: From a bioelectrical perspective, it is intuitive to hypothesize that the possible “RF current redirection effect” resulting from variations in the DP position would be more pronounced when the active and dispersive electrodes are in closer proximity”. Are you describing dispersion of energy encircling the heart away from the DP? If not, this hypothesis seems illogical. Please clarify.”

Response: The reviewer is correct in this observation. To enhance clarity, the sentence has been revised to refer explicitly to the distribution of dissipated power around the active electrode, rather than using the more ambiguous term "RF current redirection effect."

Reviewer 2 Report

Comments and Suggestions for Authors

The manuscript entitled Body Size Modulates the Impact of the Dispersive Patch Position During Radiofrequency Cardiac Ablation is an original paper. The authors assessed how individual body size may modulate the extent of the lesion size depending on the dispersive patch orientation of the ablation electrode. They used three computational models representing different body sizes.

They saw that lesion size was consistently and significantly greater with concordant dispersive patch positioning compared to discordant positioning. In addition, it may be more significant in individuals with smaller body volume.

This theme is important because the efficacy of ablation procedures remains limited in part by the inability to consistently achieve transmural lesions. In addition, there are only two clinical evidences on the effect of the dispersive patch position (where different positions were evaluated).

The manuscript is well-designed. The most interesting part is the discussions, where the hypotheses explaining the results are detailed.

The limits of the study are well defined.

Minor revision

The authors repeated two times (on page 2 and 3) the following statement : ``The only two clinical evidences on the effect of the DP position…``. I recommend decreasing the introduction which is too long. Discussions about the studies published until now with this theme is better te be included in the chapter discussions (not in introduction).

Author Response

Thank you very much for the prompt and thorough review. We have carefully addressed all comments and incorporated the corresponding revisions in the updated version of the manuscript.

COMMENT #1: The authors repeated two times (on page 2 and 3) the following statement: ``The only two clinical evidences on the effect of the DP position…``. I recommend decreasing the introduction which is too long. Discussions about the studies published until now with this theme is better to be included in the chapter discussions (not in introduction)

Response: Please note that while the two studies referenced on page 2 correspond to clinical trials, the two studies cited on page 3 are preclinical experimental investigations involving animal models. This is not a repetition but a coincidental overlap. We acknowledge that the Introduction is somewhat lengthy; however, we prefer to comprehensively review the literature regarding dispersive patch (DP) positioning during RFCA within the Introduction, reserving the Discussion section for the presentation of our findings and their comparison with existing literature.